# Transfer printing micro-assembly of silicon photonic crystal cavity arrays: beating the fabrication tolerance limit

Sean P. Bommer [1] ✉, Christopher Panuski [2], Benoit Guilhabert [1], Zhongyi Xia [1], Jack A. Smith[1], Martin D. Dawson [1], Dirk Englund [2] & Michael J. Strain [1] ✉

Photonic crystal cavities (PhCCs) can confine optical fields in ultra-small volumes, enabling efficient light-matter interactions for quantum and non-linear optics, sensing and all-optical signal processing. The inherent nano-metric tolerances of micro-fabrication platforms can induce cavity resonant wavelength shifts two-orders of magnitude larger than cavity linewidths, prohibiting fabrication of arrays of nominally identical devices. We address this device variability by fabricating PhCCs as releasable pixels that can be transferred from their native substrate to a receiver where ordered micro-assembly can overcome the inherent fabrication variance. We demonstrate the measurement, binning and transfer of 119 PhCCs in a single session, producing spatially ordered arrays of PhCCs, sorted by resonant wavelength. Further-more, the rapid in-situ measurement of the devices enables measurements of the PhCCs dynamic response to the print process for the first time, showing plastic and elastic effects in the seconds to hours range.

Two dimensional photonic crystal cavities allow the extreme confinement of light within solid state materials. Their ultra-high spatial confinement, together with resonant quality factors (Q-factors) that can exceed $10^6$ in common materials such as silicon[1,2], allow for significant enhancement of light-matter effects necessary for sensing[3,4], non-linear optics[5,6], ultra-compact lasers[7,8] and coupling with single photon emitters[9–11]. By using a suspended device geometry[12] the optical mode can be strongly confined to the semiconductor material with modal confinement provided by arrays of etched holes in the plane of the cavity, and high refractive index contrast with air above and below the device. For devices with resonant wavelengths in the telecommunications spectral band, the etch hole dimensions are on the order of 150 nm in diameter, through a suspended membrane thickness of 220 nm, ensuring a single supported optical mode in the vertical dimension.

The ultra-high optical confinement of these devices, and the sub-micron scale of their critical geometric features, makes them highly susceptible to variations in the fabrication process. Although PhCCs have been demonstrated using deep UV lithography methods[13–15], the highest level of accuracy in defining their physical geometry can be obtained by using electron beam lithography, where typical fabrication variances are in the few nanometre range, after both lithography and etching stages[16,17]. The physical variation in these critical features can in turn lead to variation in the device resonant wavelength that can be in the few nanometre range, within the 1550 nm spectral region[18,19]. For devices with linewidths that can be over 2 orders of magnitude less than this variation, direct fabrication of nominally identical devices becomes impossible. In many applications, use of a single resonator, or identification of 'hero performance' is sufficient, and so particular devices can be selected from an array where properties will vary. However, as PhCCs begin to be applied in cases where ensembles[20], or coupling between co-resonant cavities are required, it is imperative to move beyond the fabrication limitations of the platform. One recent approach employed laser enhanced selective oxidation of an array of cavities after initial spectral characterisation to align the resonant wavelength of 64 PhCCs to within a standard deviation of 2.5 pm[20]. This actively monitored trimming process allowed an ensemble of

[1]Institute of Photonics, Dept. of Physics, University of Strathclyde, Glasgow, UK. [2]Research Laboratory of Electronics, MIT, Cambridge, USA. ✉e-mail: sean.bommer.2014@uni.strath.ac.uk; michael.strain@strath.ac.uk

devices to operate on a single optical pump beam for high speed spatial light modulation. Post-fabrication tuning of individual PhCCs has also been demonstrated using mechanical[11,21], cladding refractive index[22,23], thermal[24] and electronic[25] tuning mechanisms, though routes to scaling these for dense arrays of cavities are challenging.

In this work we present a physical transfer method where, rather than tuning cavities in a fixed spatial arrangement, the individual devices are fabricated as mechanically separable pixels which can then be characterised, binned and physically rearranged onto a new substrate. This represents two major challenges in the creation of arrays fabricated from single pixel devices. Firstly, the PhCC pixel devices must be detachable from their native substrate and transferred onto a receiver substrate, whilst maintaining a suspended geometry to preserve their optical characteristics. Secondly, the process of optically characterising the devices needs to be carried out within the same system as the transfer process to enable high-throughput of device binning and assembly. This latter point is critical as the majority of high-accuracy transfer techniques rely on physically shuttling samples between characterisation and micro-assembly systems. Here, by integrating both measurement and transfer assembly processes into a single system, we not only enable the deterministic selection and transfer of 119 devices in a single session, but also unlock the possibility of measuring dynamic changes in device performance during the printing process that are inaccessible using traditional serial integration and measurement methods.

Here we measure and transfer 119 silicon PhCC devices into an array ordered by resonant wavelength. We demonstrate that the transfer printing process is repeatable, successfully repositioning single-pixel devices up to five times before their resonant wavelength shifts beyond their original resonant linewidth. We integrate a swept-wavelength spectral measurement system with the transfer printing tool to enable in-situ optical characterisation. Using this setup, we observe dynamic effects of the printing process on the cavities' resonant wavelengths over timescales from seconds to hours, uncovering elastic effects during transfer.

## Results
### Transfer printing of PhCCs and in-situ spectral measurement
The process for transfer printing of micron-scale membrane photonic devices is well established[26–29], and typically involves the use of a soft polymer stamp to pick-up, align and place devices onto host substrates. The vast majority of work has focussed on the printing of membrane devices onto planar substrates where contact is made across the full membrane surface[4,30–33]. Transfer of devices with surface contact dimensions in the few micron range have been demonstrated with high-yield and repeatability for full surface contact of micro-LED pixels[34,35], micro-lenses[36], and nanowires[37]. Successful transfer printing has also been demonstrated for devices where the contact area of printed devices is in excess of any suspended regions or areas with topological distortion.[38–40] For PhCC devices, the membranes must be printed with a suspended geometry to preserve the air-cladding above and below the PhCC for strong optical confinement, and replication of performance on its native substrate. Figure 1 shows a schematic and scanning electron microscope images of our single pixel PhCC devices. The optical cavity needs to be large enough to maintain the optical confinement of the mode, whilst minimising the total area to enable multiplexing of pixels together on a host substrate. Furthermore, when releasing micron-scale membrane devices from their host substrate, the physical support structure, or tether, has to be carefully designed to allow cleaving and prevent collapse of the membrane onto its substrate. In this work, the silicon PhCC devices used follow the design of an L3 cavity based on references[20,41]. The full photonic crystal area is $7 \times 9\ \mu m^2$. To minimise the total pixel area, and achieve high density integration, the external planar membrane border around the PhCC was limited to 1 μm in width. Figure 1a shows a schematic of the printed PhCC in its suspended geometry. In this work, the PhCC devices were printed directly onto the support structures with no additional adhesion layers. The lateral overlap between the supporting rectangular frame and PhCC border region of ≈1 μm, requiring a transfer print positioning accuracy in the 100's of nanometres range. Prior work within the group has demonstrated this level of accuracy to be achievable by use of microscopy overlay alignment methods[42,43]. By selecting fiducial markers with non-periodic geometry, it is possible to use correlation techniques to achieve sub-pixel alignment accuracy during device transfer. Furthermore, by transferring the PhCC devices onto a receiver structure, the separation of the membrane bottom surface and underlying substrate is no longer bound by the format of the fabrication wafer. In suspended silicon PhCC devices, a 220 nm silicon core is suspended over a silicon substrate at a distance of 2 μm, matching the thickness of buried oxide that is chemically etched, after lithography and reactive ion etching of the PhCC holes structures, to create an under-cladding of air. Figure 1b shows a scanning electron

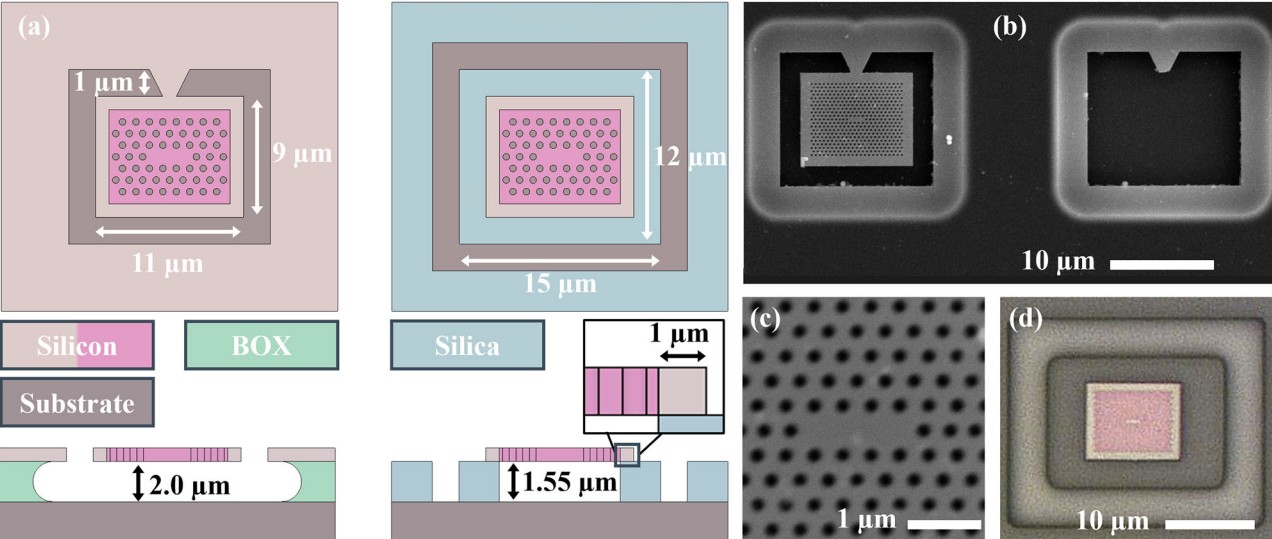

**Fig. 1 | Releasable PhCC pixels for mechanical transfer. a** Schematic of a PhCC pixel on its donor substrate and after transfer to a receiver substrate with silica support frame. **b** SEM image of fabricated PhCC pixels on the donor substrate, where the right hand pixel has been transferred, leaving a void in the donor array. **c** High magnification SEM image of the PhCC central region showing the L3 cavity geometry and (**d**) optical microscope image of a silicon PhCC printed onto a silica support frame.

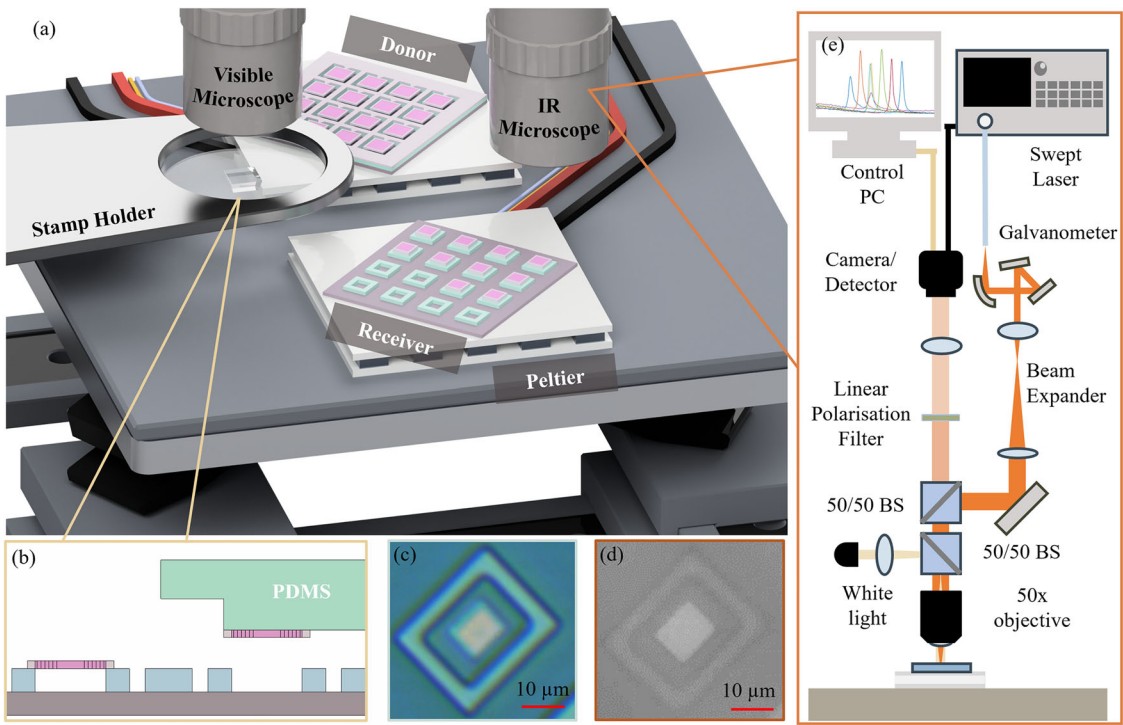

**Fig. 2 | Accurate transfer printing system with in-situ reflectivity spectrum measurement capability. a** Schematic transfer printing system and optical measurement rig incorporating high-accuracy 6-axis stage, fixed stamp holder, Peltier mounted donor and receiver samples and optical microscope objective lenses. **b** Schematic of the pixel printing process using a PDMS stamp. **c, d** show images of a printed PhCC pixel in the visible and IR systems respectively. scale bars represent 10 μm. **e** Schematic detail of the optical injection and measurement system embedded in the transfer printing tool.

microscopy (SEM) imaging of a representative PhCC device suspended from the native silicon substrate (left) beside a void where a PhCC has been transferred (right). Figure 1c is an SEM image close up of the representative L3 cavity of a suspended PhCC on the native silicon substrate. In the receivers fabricated for this work, we used Plasma Enhanced Chemical Vapour Deposition (PECVD) silica layers of 1.55 μm thickness on a silicon substrate. The silica suspension frames were fabricated using optical lithography and reactive ion etching. The 1.55 μm separation was deliberately targeted to promote constructive interference between the internal PhCC mode and the reflected light from the air-substrate interface, to improve vertical out-coupling of the cavity towards the microscope measurement optics[44]. An example of a printed PhCC on a receiver silica frame in suspended geometry is shown in Fig. 1d.

A schematic of the transfer print system with optical measurement module is presented in Fig. 2. The transfer printing system comprises an optical microscope column, fixed transfer stamp holder and a high-accuracy 6-axis translation stage with Peltier coolers, on which the donor and receiver samples are mounted[45]. The Peltier elements are held a few degrees below the cleanroom laboratory temperature at a constant 19.6 °C with closed loop feedback to ensure stable operation of the PhCCs. In addition to the wide-field optical imaging function of the microscope column, the system also incorporates an optical injection line for the swept-wavelength tuneable laser characterisation system. The signal reflected from a targeted PhCC is coupled back through the microscope objective to a camera system for assessment of the spatial mode and measurement of the reflectivity spectrum under swept wavelength operation.

Figure 3 shows a reflectivity spectrum measurement from a representative PhCC device captured using the in-situ scanning laser measurement system in the transfer print tool. The central resonant wavelength, linewidth and Q-factor of the resonator can be extracted from this swept laser measurement in real-time, with single pass

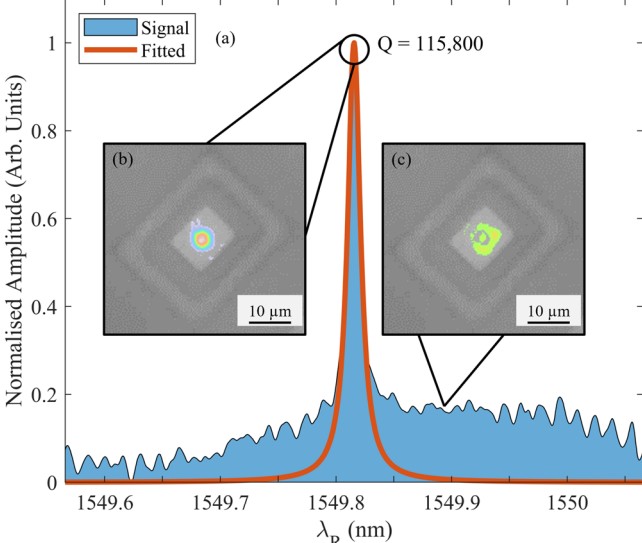

**Fig. 3 | Spectral measurement of PhCC in the transfer system. a** Measured reflectivity spectrum from a PhCC captured using the in-situ measurement system. The PhCC is printed on a receiver substrate silica suspension frame. Insets show the spatial mode images captured by the InGaAs camera at two points in the tuneable wavelength sweep corresponding to on-resonance (**b**) and off-resonance (**c**) conditions.

sweeps taking only a few seconds to complete. Details of the method for parameter extraction from the spectral measurements is presented in the supplementary note 2.3. For the PhCC's in this work, the resonant wavelengths were in a range around ~1551 nm, with cavity linewidths of ~15 pm and Q-factors in the $10^5$ range.

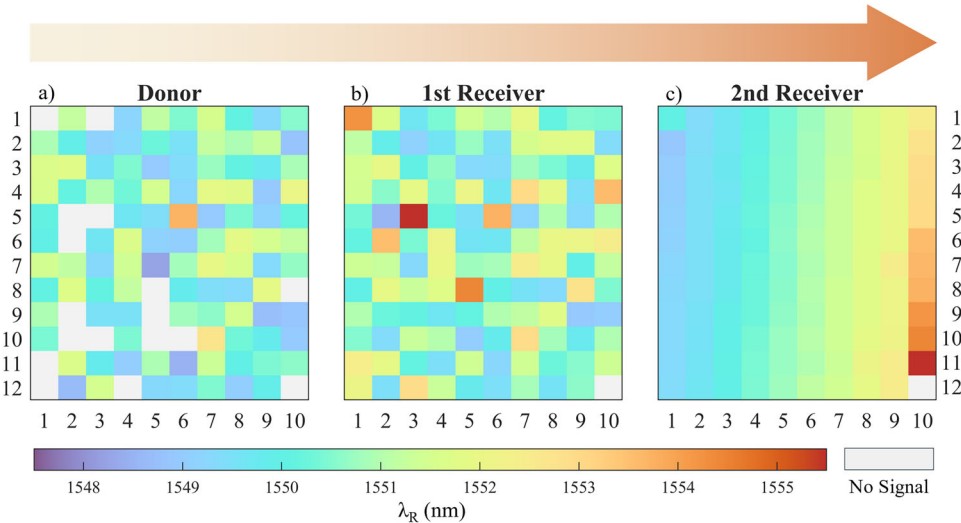

**Fig. 4 | Spatial ordering of PhCC arrays by resonant wavelength.** Measured resonant wavelengths of the PhCCs on (**a**) donor substrate, (**b**) 1st receiver and (**c**) 2nd receiver substrate.

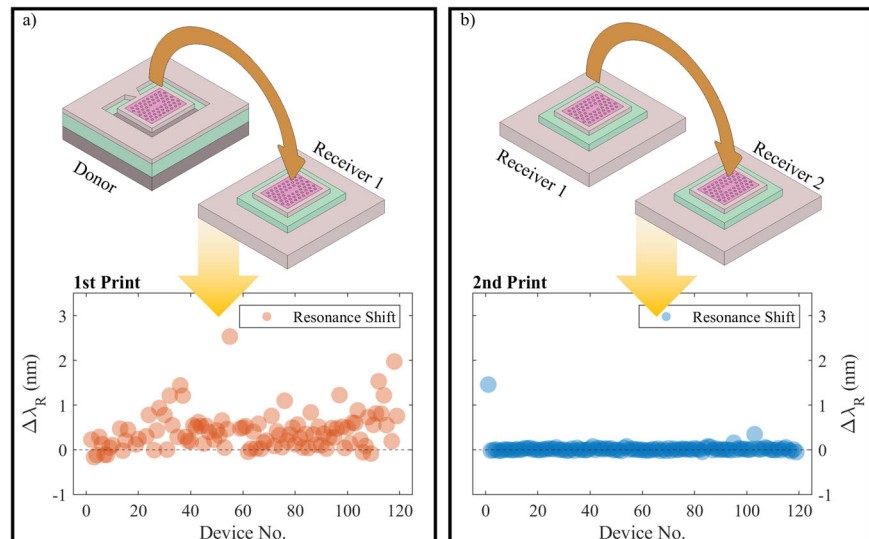

**Fig. 5 | Printing induced resonant wavelength shift.** Measured resonant wavelength shifts of individual PhCCs after (**a**) 1st print, and (**b**) 2nd print.

## Spatial ordering of PhCC array by resonant wavelength

To enable spatial ordering of an as-fabricated PhCC array, 120 devices were first measured on their native substrate, and the resultant map of PhCC pixel resonant wavelengths is shown in Fig. 4a. As expected, the measured cavity wavelengths span across a range of a few nanometres in wavelength, corresponding to the variations in cavity geometry induced by nano-fabrication geometry tolerances. The blank spots in the figure correspond to PhCCs where no spectral measurements could be obtained, likely due to partial collapse of the pixels onto the exposed substrate during the under-etch process. One pixel could not be released from the donor chip, so all further results correspond to the remaining 119 pixels that could be transferred. The resonant wavelengths are distributed across the array without any clear spatial pattern, as shown in Fig. 4a. The devices were then numerically sorted by resonant wavelength and printed onto receiver substrate following the spectral ordering.

Figure 4b shows a map of the printed pixels' resonant wavelengths on a first receiver substrate. From the first print it is obvious that the spectral ordering has been frustrated and again the cavities form a disordered array, although all transferred pixels are now visible,

recovering the previously unmeasureable cases from the donor substrate. The measurements on receiver 1 were then used to create an ordered array, through selective placement during an additional transfer step onto a second receiver, the measured results of which are presented in Fig. 4c. The wavelength ordering was preserved faithfully in this transfer, indicating that the process of cleaving the silicon tethers during the first print induced a plastic shift in the cavity resonant wavelengths. Figure 5a, b show the wavelength shifts of individual PhCCs for the first and second prints, with a clear improvement in the second print case. For the first print, the mean wavelength shift of ± 0.426 nm and standard deviation across the array of ± 0.438 nm, clearly shows the effect of the release from the donor substrate on the cavities where the spectral ordering has been frustrated by the initial release process. The second print however shows a mean wavelength shift of ± 0.025 nm and standard deviation of ± 0.139 nm, demonstrating stable performance of the PhCCs before and after the print comparable with what can be achieved with post-fabrication trimming. The two outlier values of wavelength shift in the second print array correspond to two cases where mechanical effects during printing caused larger deviations of the resonant wavelength. Device 1 was

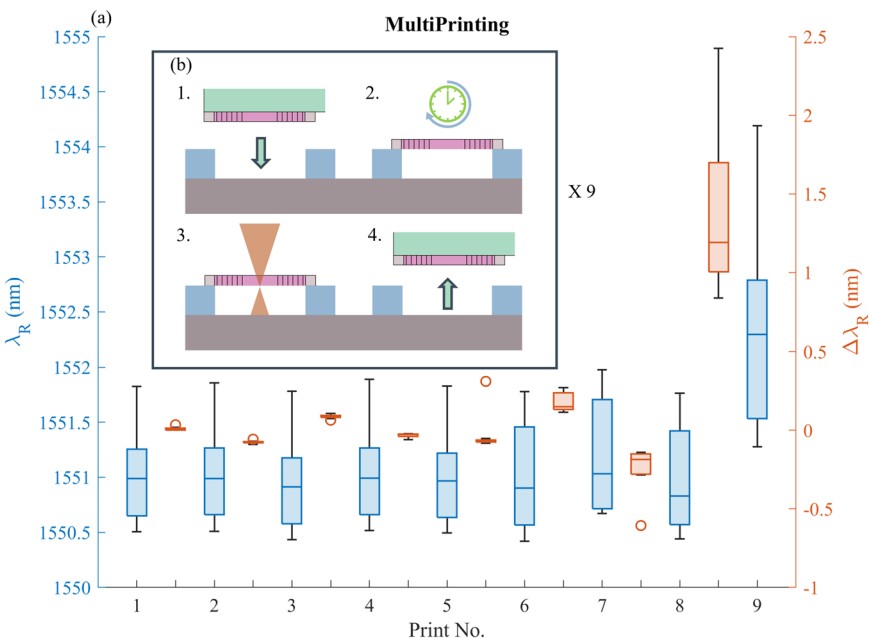

**Fig. 6 | Repeated print cycles. a** Measured cavity wavelength sets after each printing cycle for a set of 10 PhCC devices, showing absolute cavity resonant wavelength (blue) and relative resonance shift between each print location (red). Box plots indicate median value (middle line), 25th, 75th percentile (box) and 5th and 95th percentile (whiskers) as well as outliers (single points). **b** Schematic of the print and measurement cycle.

printed as a calibration step for the force being applied during the print release stage and a higher than necessary value appeared to cause a residual plastic shift in resonant wavelength. The second outlier device was accidentally printed onto a rigid area of the sample before being moved to its target receiver location, again exhibiting a larger deviation of the resonant wavelength. Removing these outliers in the second print gives mean and standard deviation wavelength shifts of ± 0.007 nm and ± 0.021 nm respectively. By spatially ordering the devices, local clusters of PhCCs with resonant wavelength spread within a cavity linewidth can be achieved, without post-fabrication trimming. Examples of cavity clusters are presented in the supplementary note 3.

An additional set of 10 PhCCs from the donor chip, with resonant wavelength around 1550 nm were selected to test the effects of multiple print cycles on the cavity response. Figure 6 shows the measured resonant wavelengths of the PhCCs after each print cycle and the relative shift of individual devices. All print cycles are numbered after the initial print '0' from the donor to receiver to avoid inclusion of the plastic shift effects in these results. For 5 print cycles, after the initial print from the donor substrate, the subset of PhCCs show negligible wavelength shifts. The stability then degrades over the next few cycles, reverting to deviations close to the as-fabricated array. This could be due to the effects of surface contamination from the environment or from the transfer stamp itself, but no clear effects were visible under electron microscope inspection.

The wavelength ordered set of PhCCs presented above was printed with minimum pitch of 50 μm in both the x and y directions. In order to demonstrate the potential of this integration method for producing densely packed arrays, we transferred an additional 120 devices onto an 8 × 15 spatial array of support frames with a pitch of 18 and 16 μm in the x and y directions and with a minimum edge to edge separation of 7 μm in both directions, as shown in Fig. 7.

## Dynamic cavity response measured in-situ

As detailed in the previous section, the release of the silicon PhCC pixels from their native substrate results in a permanent shift in their cavity resonant wavelength. Since the spectral measurements of the PhCCs are carried out in-situ in the transfer printing system, dynamic

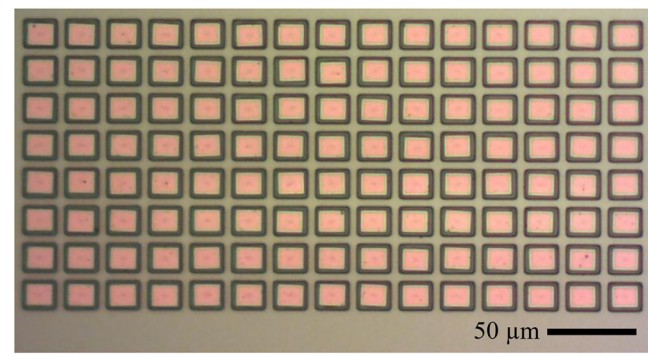

**Fig. 7 | Densely printed array.** Optical microscopy image of an an array of 120 PhCC devices printed with edge to edge spacing of 7 μm in both directions.

effects can be monitored, limited by the timescale of the swept laser spectral measurement system. The inset to Figure 8 shows the time varying cavity reflectivity spectrum for a single PhCC cavity over a few tens of seconds just after the initial printing, showing a clear shift of the spectrum with time. The main decay curve presented in Fig. 8 shows the results of spectral measurements of 2 different PhCCs on the same substrate measured concurrently over the course of 200 minutes, plotting the measured resonant wavelength shift relative to the pre-printed steady-state value. The printing process shows a clear red-shift in cavity resonant wavelength immediately after release onto the receiver substrate, followed by a relaxation back towards the steady state value on a timescale of around 200 minutes.

The results of the multiple print cycles show that steady-state cavity resonances are preserved for up to 5 cycles, demonstrating that the print process does not leave measurable material deposits on the cavity, which would induce a red-shift of the resonance wavelength due to increasing local cladding refractive index. The dynamic relaxation of resonant wavelengths rather suggests a strain based effect induced by the contact printing that then relaxes as the free-standing PhCCs are left on the suspension frames with no restraining force beyond the local forces between their surfaces. Furthermore, the relaxation curves can be fitted to a double exponential function, with

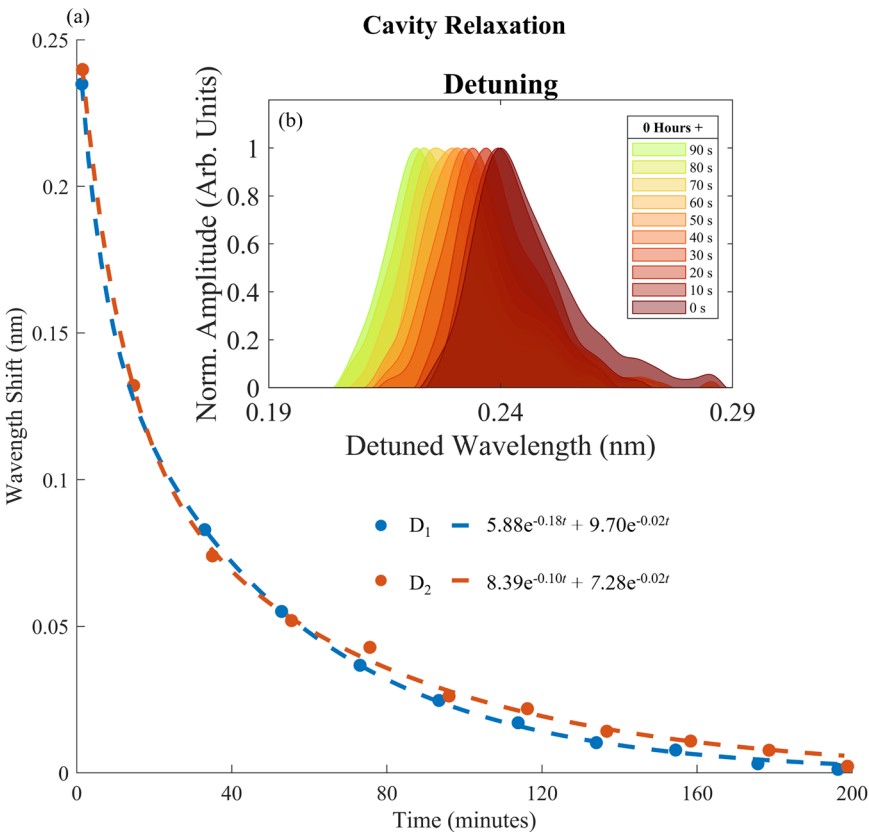

**Fig. 8 | Dynamic effects of printing on the cavity optical response. a** Cavity resonance wavelength for 2 separate PhCCs measured over a 200 minute period after initial printing. **b** The fast relaxation of the cavity resonance just after printing, as captured by the in-situ measurement and the stable steady state response after relaxation.

two characteristic relaxation time constants originating from potentially distinct physical processes.

## Discussion

The integration of spectral measurement and transfer print integration into a single system not only benefits the throughput, yield and selectivity of the device integration process, but has also enabled measurement of transient effects in device performance that were not possible in separate integration and measurement systems. Furthermore, the integration of high quality factor, air clad, photonic crystal cavities provides an extremely sensitive optical measurement tool for observing these transient effects. As noted above, there are two measurable effects of the PhCC integration process on the cavity resonances. The first is a permanent shift in the cavity resonant wavelength during the release from the donor substrate, and the second is a transient resonant wavelength shift during subsequent prints that relaxes to a consistent, steady-state value. Although full exploration of the physical mechanisms underlying these effects is beyond the scope of this work, their magnitude and potential sources can be outlined.

A shift of PhCC resonant wavelength measured after a print cycle can be due to change of cavity effective refractive index, or mechanical deformation of the cavity. In this work we have used nominally identical L3 photonic crystal cavities that can have measured resonant wavelengths with a standard deviation in the order of 1 nm (around the 1550 nm region). In reference[2] the authors show that a wavelength dispersion of this order corresponds to a randomised variation in hole radius and position of $\approx 0.0014a$, where $a$ is the lattice constant of the photonic crystal. This in turn would correspond to a variation in real space of $\approx 0.56$ nm (for a lattice constant of 400 nm) for our devices. Thus, in accordance with perturbation theory[46], a uniform expansion of the PhCC lattice by 0.56 nm would lead to a redshift of cavity wavelength greater than 1 nm. Similarly, given the near-unity energy

confinement of our L3 cavity modes in silicon, perturbation theory also dictates that a 1 nm redshift in wavelength corresponds to a change in effective index of a factor of $\approx 0.0014$, assuming no mechanical deformation of the cavity. In reference[20] oxidation of silicon PhCC's show resonant wavelength blueshifts of a few nanometres, through oxide formation of nanometric scales. Thus, effective refractive index changes of this order correspond to nanometric material refractive index changes. We are unable to measure any deposition of stamp material on the cavity via direct imaging (optical or electron beam) that would show such nanometric depositions. In the case of the transient resonance shifts from one suspended position to another, the cavity response relaxes to a consistent steady state, within the linewidth of the resonance. This suggests that any physical change in cavity effective index or mechanical stress, relaxes over the measured time period. This in turn suggests a stress induced in the cavity due to printing that then relaxes, enabled by the surface to surface contact between the membrane and support structures.

The sub-nanometre wavelength shifts of all cavity steady-state resonances observed after 5 sequential printing cycles demonstrates the ability to preserve all physical properties of the device (lattice constant, refractive index, etc.) to better than one part in 1000. The preservation of resonant cavity wavelength allows for the selection of specific cavities to be integrated into pre-defined spatial locations, overcoming the existing variance in nominally identical devices as-fabricated on their native substrate. This deterministic post-measurement selection of cavities will enable single device selection for integration into photonic integrated circuits, or on fiber tips, as high-quality factor filters and non-linear resonators with absolute control over target wavelength. Importantly, the spatial ordering of these devices enables construction of cavity arrays with well-defined wavelength characteristics. For example, cavity arrays can be designed with wavelength gradients across the array to enable passive beam

steering at high speeds[47]. Uniform arrays can also be assembled for applications as spatial light modulators[20]. In this case the yield of cavity responses within the linewidth scale is important and is related to the variance of cavity wavelength from nanofabrication tolerances. In this work we show that groupings of $\approx 5$ PhCCs can be extracted within the range of a linewidth from an initial set of 120 devices. Full details are presented in the supplementary note 3. This gives a nominal down selection factor of $\approx 0.042$, meaning that for a uniform $10 \times 10$ cavity array, an initial donor set of around 2380 devices is needed. Complementarily, for donor arrays at large scale, multiple uniform arrays can be assembled from a single fabrication run.

We have demonstrated a post-fabrication integration method that allows the spectral measurement, binning and spatial arrangement of high Q-factor Photonic Crystal Cavity (PhCC) devices into ordered arrays, with precision beyond what is achievable in as-fabricated device arrays. The transfer printing integration of 119 PhCC pixels demonstrated handling of devices with dimensions in the few micrometre range, onto suspension frames on receiver substrates with physical contact limited to $1\,\mu m$ in overlap. SEM imaging was used to characterise the physical overlap of the PhCCs and support frames, as presented in the supplementary note 1.2. The printing process preserved cavity resonant wavelengths within their linewidth range after a first permanent shift induced by the print release from the donor substrate. Furthermore, the printing process preserved cavity performance over multiple cycles, allowing for reconfiguration or rework of samples using this method. The in-situ optical measurement of cavities enabled study of the printing dynamics in the seconds to hours timescale, exhibiting elastic relaxation effects in the cavities after printing that would not have been easily measurable in standard integration and measurement system combinations. The ability to directly measure micron scale device response and subsequently integrate large numbers of devices onto host chips breaks the dependence on fabrication limited integrated optical systems and paves the way for future, high performance optical systems-on-a-chip.

## Methods

### Photonic crystal cavity devices
The PhCCs used in this work were fabricated on a native 220 nm core thickness Silicon-on-Insulator (SOI) platform, with a $2\,\mu m$ buried oxide layer between the core and silicon substrate, following the design of cavities from references[20,41]. The PhCC cavities were defined on releasable pixels, where the PhCC was surrounded by a $1\,\mu m$ wide planar silicon area and a concentric trench area with a width of $1\,\mu m$. The pixel geometry is shown in Fig. 1. The PhCCs were suspended, and released from the substrate using a chemical HF acid etch to selectively remove the buried oxide layer. They were tethered to the surrounding core area during under-etching using a tapered section of silicon core that supported the pixels in a cantilever geometry during suspension. The transfer printing process cleaved this tether during pick-up to release the PhCCs from their native substrate.

### Receiver substrate fabrication
The receiver substrate for the transfer print based integration of PhCCs required a geometry matching the lateral overlap dimensions of the PhCC pixels and a vertical suspension over a substrate. To match the donor geometry of silicon membranes suspended above a silicon substrate, a silicon substrate was selected as a receiver. The suspension frames were fabricated in silica, again matching PhCC geometry, with a height of $1.55\,\mu m$. The silica layer was deposited onto the silicon substrate using a Plasma Enhanced Chemical Vapor Deposition (PECVD) process. The suspension frames were then defined using direct write laser lithography into a photoresist layer spin-coated onto the silica, then transferred to the silica using $CHF_3$ Reactive Ion Etching (RIE). Finally, the photoresist layer was removed using organic solvents and a short $O_2$ plasma ash.

### Transfer print system and in-situ optical measurement
The transfer printing system was a custom built instrument comprising a 6-axis motion control stage, rigidly fixed stamp holder, polymer (PDMS) stamp pick-up head, optical microscopy system and custom control and alignment software. Details of this system can be found in reference[45]. The donor and receiver samples were mounted on the motion control stage and their positions measured using the optical microscopy systems and on-chip alignment markers. The PDMS stamp was aligned to a PhCC pixel and brought into contact to enable pick-up using a fast retraction step. The selected PhCC pixel was then spatially aligned to the receiver chip structures and brought into contact with the suspension frame. A slow retraction step then leaves the PhCC pixel deposited on the receiver substrate.

The optical measurement system consisted of an Agilent swept laser setup fibre coupled to a 2 axis galvanometer system to allow scanning of the beam spot across the microscope field of view. The reflected beam spot was coupled to a Vis-SWIR InGaAs camera for spectral measurements using time domain sampling of the spot intensity. The combination of the visible light widefield illumination for optical imaging of the sample, with the scanning IR beam spot measurement line, enabled simple spatial alignment of the injected laser source with the PhCCs on-chip. The injected and reflected laser beam paths were prepared in a cross-polarised arrangement to isolate direct specular reflection from the sample surface. Further details of this measurement system are presented in the supplementary note 2.1.

### Optical spectra and dynamic response measurements
The reflection spectra from the PhCCs was extracted by summing the reflected beam intensity across a region of interest in the InGaAs camera. Each camera frame was taken as a single time sample, correlated with an instantaneous laser wavelength during the continuous sweep cycle. The laser was swept at a rate of 0.5 nm/s and the camera frame rate was 833.3 fps giving a sampled spectral resolution of 0.6 pm. For the measurements of device arrays the laser was swept over a full span of 10 nm giving a measurement time of 20 s, enabling coverage of the full span of resonant cavity wavelengths across the array. Cavity resonant wavelength and Q-factor were calculated using peak search and resonance fitting routines in software at the end of each scan. For the dynamic measurements on pre-characterised devices the laser scan range was reduced to 2.5 nm, reducing the sweep time to 5 s, and giving a total scan and analysis time of less than 10 s.

## Data availability

The spectral data generated in this study have been deposited in the figshare database under accession code https://doi.org/10.6084/m9.figshare.28280786.

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

## Acknowledgements

The authors acknowledge funding from the following sources: Royal Academy of Engineering (Research Chairs and Senior Research Fellowships); Engineering and Physical Sciences Research Council (EP/R03480X/1, EP/V004859/1); Innovate UK (50414). Authors also thank Dr. Eleni Margariti and Elise Burns for their contributions to the review process in terms of additional fabrication and analysis.

## Author contributions

S.P.B., M.D.D. and M.J.S. conceived the idea. S.P.B. and M.J.S. led the research. C.P. and D.E. led the PhCC cavity design and microfabrication. S.P.B. and B.G. designed and built the optical injection and transfer system. S.P.B., B.G. and Z.X. led on micro-transfer printing and

microfabrication and J.A.S. led on automated resonance analysis. S.P.B. and M.J.S. wrote the manuscript with input from all authors.

## Competing interests

The authors declare no competing interests.
