## [Transparent Peer Review file · Nature Communications]

Transfer printing micro-assembly of silicon photonic crystal cavity arrays: beating the fabrication tolerance limit

Corresponding Author: Mr sean bommer

Version 0:

Reviewer comments:

Reviewer #1

(Remarks to the Author)

The authors introduced a transfer printing method combined with in-situ optical measurement to achieve a highly ordered PhCCs array with minimal performance variation. The rapid in-situ measurement facilitates the binning and transfer of PhCCs and allows monitoring of the dynamic variation in the cavity optical response of the printed device. This manuscript may be considered for publication if the authors sufficiently address the following comments and suggestions.

Comments and suggestions

Does the protruding PDMS stamp used to pick up the suspended Si PhCCs have a flat surface, or is it micro-structured for adhesion switching? Could the authors provide some microscopic images of PDMS stamp including size and more detailed stamping procedure in the manuscript?

Transfer printing of Si membrane using PDMS is quite common method. Could the authors address the novelty of transfer printing technique used here, compared to other papers?

In Method section, the authors mentioned that fast and slow retractions are required to pick-up and release the device. How specifically fast should these two processes be carried out?

Could the authors discuss potential issues that might happen during the transfer process, such as mechanical cracks or deformation (e.g., sagging) of the Si PhCCs, and residual surface charge effects? Additionally, how can these issues be prevented?

Please provide the images or data that can clearly demonstrates the lateral overlap dimension between PhCC and silica suspension frame is less than $1\mu\text{m}$.

Please provide the image of the sample that shows transferred and spatially aligned 119 devices.

Please give explanations why wavelength shifts of PhCCs are alleviated after 2nd print (Fig. 5).

Please give more detailed explanation why the PhCC after transfer has red-shift of resonant wavelength, not blue-shift (Fig. 7).

Reviewer #2

(Remarks to the Author)

For fabricating PhCCs, it is extremely difficult to avoid variations in resonance wavelength and Q-value and fabricate PhCC arrays with the same performance. However, it is worth challenging to investigate technological development to solve this issue. This paper reports on the development of a technology needed toward realizing an array structure in which each PhCC has the same performance using a transfer printing method (TP), and describes the successful transfer of 199 PhCCs to produce an array of spatially aligned PhCCs classified by resonance wavelength. Furthermore, it is shown to be possible to measure the dynamic transient of PhCCs to the TP process.

However, although this paper reports remarkable results on a spatially aligned PhCC array classified by resonant wavelength using the TP process, it is far from achieving a PhCC array with the same performance such as the same Q-value and the same resonance wavelength. In addition, the technology used is based on existing laser scanning technology and TP, and the novelty of the technology is not sufficiently demonstrated. In order for this paper to be suitable for publication in Nature Communications, the following points should be clarified

(1) The authors have shown that the TP process can transfer a single pixel element up to five times before the shift in resonance wavelength exceeds the line width, but it seems that the overlap with the receiver area of less than 1 micron can be applied under special constraints such as the current size of the PhCC and the slab thickness so on. I would like to suggest the authors to show the dependence of the success rate of TP on the overlap area with the receiver area and the dependence of the PhCC slab thickness, etc.

(2) The TP equipment developed by the authors is excellent but appears to be a standard one, and it is difficult to see its originality. For example, the swept wavelength spectrum measurement system integrated with the TP equipment does indeed enable in-situ evaluation of optical properties, but it does not seem to have anything particularly new in terms of scientific development. I would like suggest the authors to emphasize the novelty of the TP device to be clearly.

(3) The dynamic effect of the TP on the resonance wavelength of the cavity is measured on a time scale of several seconds to several hours, and the elastic effect of the response is discussed, which is an interesting result, but the physics that underlies this time scale is not clear. In order to enable more comprehensive discussion, the authors should show the dependence of the closeness of the receiver and the PhCC, the overlapping area, the PhCC slab thickness, and the overall size, etc.

(4) The TP technology developed in this study is a combination of existing technologies, and its usefulness of PhCCs array as a device development has not been demonstrated, so it remains at the stage of half-baked research. The authors are expected to clarify how the array device ordered along the wavelength will be used in practice.

Version 1:

Reviewer comments:

Reviewer #1

(Remarks to the Author)

The authors have done a reasonable job in responding to referee inputs. The revised version is now suitable for publication.

Dear Editor,

We would like to thank the reviewers for their detailed consideration of our manuscript and their constructive comments and suggestions. We have addressed the reviewers' comments in the revised manuscript and supplementary files and we give full details of these changes in a point by point manner below. We believe that these modifications, and the further work carried out as suggested by the reviewers, have made substantial improvements to our manuscript.

Reviewer 1

1. *Does the protruding PDMS stamp used to pick up the suspended Si PhCCs have a flat surface, or is it micro-structured for adhesion switching? Could the authors provide some microscopic images of PDMS stamp including size and more detailed stamping procedure in the manuscript?*

The micro-fabricated PDMS stamp has a flat surface that is brought into contact with the photonic crystal cavity devices for transfer. We have now included images of the stamp in the supplementary material, along with details of its dimensions and composition. Furthermore, we agree that the detail on the transfer printing process was not presented in as much detail as it could have been. We have therefore added the following process description and additional schematic figure detailing the process to the supplementary material:

“The general method used to transfer print objects using a polymeric stamp is well covered in the literature [1–4] and typically relies on the reversible adhesion of the stamp-object interface based on competitive adhesion effects between the stamp and the donor and receiver substrates, making use of velocity control during the contact phases of the process [5]. A further variation on this basic method has been developed to reduce surface contact areas during the print phase using protrusions that can dynamically relax [6]. These methods are suitable for membranes with thicknesses greater than approximately 1 μm or with high stiffness, but the flexibility of thin device membranes, or micron-scale devices, require the use of flat stamp surfaces to avoid unwanted deformation of the devices [7]. Furthermore, the composition of the polymeric material can be engineered to control its surface adhesion to promote transfer in the required direction from donor to receiver [7]. In this work, the small-scale of the membranes and the sub-micron thickness necessitated use of a flat stamp surface, using a Polydimethylsiloxane (PDMS) composition of 10:1 (monomer:curing agent) mixture. A schematic of the transfer process is presented in Fig. S1, along with images of a PhCC attached to the polymer stamp head. The PhCC's are lithographically defined with a single, laterally tapered tether with a minimum dimension of 2 μm at the membrane edge. This tether was designed to fracture at the minimum cross-section and all 249 devices (including the ordered wavelength device set, the repeat print cycle set and the dense integration set) transferred successfully with the tether cleaving at the designed point with no detectable physical damage to the PhCC pixel. The flat stamp surface was brought into contact with the PhCC surface, ensuring no stamp contact with the tether area. Once in contact a lateral shear motion of $\approx 1 \mu\text{m}$ was applied to fracture the tether at a velocity of $\approx 8000 \mu\text{m/s}$, Fig. S1(a). The PhCC on the stamp is then aligned to the receiver chip suspension frame, centring the PhCC on the air void at the centre of the frame. The PhCC is brought into contact with the silica suspension frame, then released from the stamp surface using a combined shear and vertical motion at a velocity of $\approx 0.05 \mu\text{m/s}$. This releases the PhCC into its target position with a yield of 100% for the 408 prints carried out for this work,

i.e. 238 prints of the main cavity set, 50 prints of the repeatability tests, and 120 prints for the dense integration set.”

2. *Transfer printing of Si membrane using PDMS is quite common method. Could the authors address the novelty of transfer printing technique used here, compared to other papers?*

The reviewer is correct that PDMS based transfer printing of silicon membranes of similar thickness has been presented in the literature, as we note in our references. Nevertheless, there are a few key elements of novelty in our system, processes, and the geometry of the printed devices, over previous demonstrations in the literature. We mention these explicitly throughout the manuscript and they can be summarised as:

- Transfer printing is typically carried out for membrane devices where the full membrane surface is in contact with the receiver substrate. The optical function of the PhCC cavities require these to retain a suspended geometry with an air-cladding above and below the silicon membrane across the full surface area of the PhCC optically active area. This is the first time such suspended geometries with such low contact area have been shown to be possible for thin membrane devices.
- Our custom transfer print system is the only system that enables in-situ optical spectrum measurement of devices. This capability is critical for the rapid measurement and binning of large numbers of devices before printing, and assessment after printing. We achieved the characterisation, transfer and re-measurement of our 119 devices within the space of a few hours due to this in-situ capability and the automated motion control system for rapid scanning across the device arrays. Although spectral measurements, and transfer printing (of non-suspended devices) have been previously demonstrated individually, their combination into a single system, along with automated device scanning and alignment represents a non-trivial advance in technology and enables the demonstrations presented in this work that would otherwise have been prohibitively time consuming to achieve using stand-alone systems.
- The rapid transition between measurement and printing modes of the tool enable measurements of the printing dynamics down to seconds timescales that would be impossible with existing equipment and separate integration and measurement systems. This capability enabled the measurement of the membrane relaxation effects after printing as detailed in the final section of the manuscript.
- The 6-degree of freedom motion control system used in our transfer printing tool has a range of velocity control from $0.01\text{-}10^3 \mu\text{ms}^{-1}$ that together with vectorial motion path allows for the challenging suspended geometry printing demonstrated here.
- The use of compact, high-quality factor photonic crystal cavities in a suspended geometry provides an incredibly sensitive probe of high spatial frequency (i.e. micron scale) physical device characteristics that have been previous inaccessible. As detailed in response to point 4 below, the unique combination of high sensitivity and small mode volume in these devices enables the most accurate characterization of membrane transfer printing physics to date.

We have amended the abstract to better address these points:

“Photonic crystal cavities (PhCCs) can confine optical fields in ultra-small volumes, enabling efficient light-matter interactions for quantum and non-linear optics, sensing and all-optical signal processing. However, the inherent nanometric tolerances of micro-fabrication platforms can induce cavity resonant wavelength shifts several-orders of magnitude larger than cavity

linewidths, prohibiting fabrication of arrays of nominally identical devices. We address this device variability by fabricating PhCCs as releasable pixels that can be transferred from their native substrate to a receiver, retaining their suspended geometry, where ordered micro-assembly can overcome the inherent fabrication variance. We demonstrate the measurement, binning and transfer of 119 PhCCs in a single session, producing spatially ordered arrays of PhCCs, sorted by resonant wavelength. Furthermore, the development of an integrated spectral measurement and transfer printing instrument enables the rapid in-situ measurement of the high Q-factor devices for the first time. This capability was used to probe the dynamic response of membrane devices to the print process for the first time, showing plastic and elastic effects in the seconds to hours range”

3. In Method section, the authors mentioned that fast and slow retractions are required to pick-up and release the device. How specifically fast should these two processes be carried out?

As detailed in our response to point 1 above, a relative velocity of 8000 $\mu\text{m/s}$ was used to break the PhCC tether and a 0.05 $\mu\text{m/s}$ motion was used to release the devices onto the silica suspension frames. Furthermore, in addition to the z-motions perpendicular to the sample surface, we also make use of shear motion in-plane with the surface. These details have now been added to the manuscript documents.

4. Could the authors discuss potential issues that might happen during the transfer process, such as mechanical cracks or deformation (e.g., sagging) of the Si PhCCs, and residual surface charge effects? Additionally, how can these issues be prevented?

As noted in our answer to point 1 above, for this work we released 249 devices from their fabrication substrate (119 in the main device set, 10 devices for the repeatability tests, and 120 devices for the dense integration demonstration) by creating a mechanical fracture of the tether at its narrowest point. The single device that was not released from the array of 120 devices from the main results had previously collapsed during the preceding release process and was therefore bound to the substrate by stiction forces. This single failure was therefore fabrication related rather than a by-product of the transfer process. These results quantitatively demonstrate the robust nature of the silicon pixel micro-fabrication process, especially considering that the devices were air-shipped between collaborator sites across the world prior to the transfer printing.

Furthermore, there was no visible mechanical damage to the PhCC pixels after any of the printing processes in terms of cracks or visible degradation of the membrane devices. The measurement of the PhCC resonant wavelength is more sensitive to mechanical/refractive index variations in the cavity than microscopy methods and is therefore reported in the main results of the manuscript as the primary mechanism for probing the condition of the PhCC devices before and after printing, for comparison.

A shift of PhCC resonant wavelength measured after a print cycle can be due to change of cavity effective refractive index, or mechanical deformation of the cavity. We have now quantified this explicitly in the manuscript to make the low impact effects of the printing process clearer, and the possible magnitude of effects from the first print as the pixels are released from their fabrication substrate. To summarize the detailed account below, the sub-nanometer wavelength shifts of all cavity resonances observed after 5 sequential printing cycles

demonstrates the ability to preserve all physical properties of the device (lattice constant, refractive index, etc.) to better than one part in 1000.

Specifically, as highlighted in the manuscript, nominally identical L3 photonic crystal cavities can have measured resonant wavelengths with a standard deviation in the order of 1nm (around the 1550nm region), which is comparable with our results. In reference [Lai et al., APL, 104, 2014] the authors show that a wavelength dispersion of this order corresponds to a randomised variation in hole radius and position of $\sim 0.0014a$, where a is the lattice constant of the photonic crystal. This in turn would correspond to a variation in real space of $\sim 0.56\text{nm}$ (for a lattice constant of 400nm) for our devices. Thus, in accordance with perturbation theory [Joannopoulos et al., *Photonic Crystals: Molding the Flow of Light*, 2008], a uniform expansion of the PhCC lattice by 0.56nm would lead to a redshift of cavity wavelength greater than 1nm. Similarly, given the near-unity energy confinement of our L3 cavity modes in silicon, perturbation theory also dictates that a 1nm redshift in wavelength corresponds to a change in effective index of a factor of ~ 0.0014 , assuming no mechanical deformation of the cavity. In reference [Panuski et al., *Nature Photonics*, 16, 2022] oxidation of silicon PhCC's show resonant wavelength blueshifts of a few nanometres, through oxide formation of nanometric scales. Thus, effective refractive index changes of this order correspond to nanometric material refractive index changes. We are unable to measure any deposition of stamp material on the cavity via direct imaging (optical or electron beam) that would show such nanometric depositions.

We have added this detail to the new discussion section in the manuscript.

5. *Please provide the images or data that can clearly demonstrates the lateral overlap dimension between PhCC and silica suspension frame is less than 1 μm .*

To quantify the spatial overlap of the PhCC devices with the support frames we carried out a new set of measurements where a support frame was imaged in an SEM system before and after printing of a PhCC device. Spatial measurements on the frame and PhCC dimensions using the SEM images were used to quantify the overlap of the devices. We have added the following text to the supplementary information along with the SEM images to detail this result.

“To quantify the extent of the spatial overlap between the printed PhCC devices and support frames, a frame was imaged in a scanning electron microscope before and after printing of the PhCC, as shown in Fig. S2. The internal dimensions of the support frame were measured as $8.5 \times 7.5 \mu\text{m}$ prior to the PhCC printing. The PhCC device was measured with external dimensions of $11 \times 9 \mu\text{m}$, and by image overlap analysis of the pre- and post-printing images the mean overlap lengths between PhCC and support frame were 1.21, 1.24, 0.49, 1.05 μm for the left, right, top and bottom edges respectively resulting in an average overlap of 1.00 μm . The edge detection was performed within Python using standard techniques within the OpenCV package, using the following process. A Gaussian blur was used to de-noise the image, following a morphological close operation to remove pixel islands that were unattached to the target structures. Finally, an adaptive threshold was used to binarize the image. An averaged line profile was taken for each edge of the imaged structures, in both the horizontal and vertical directions and passed through a peak detection algorithm from the SciPy library to find the locations of each edge, and the errors are taken as the FWHM of each imaged structure edge.”

6. *Please provide the image of the sample that shows transferred and spatially aligned 119 devices.*

Our characterised PhCC array was printed with a minimum pitch of 50 μm in both the x and y directions. The reviewer's comment prompted us to attempt to print an array with a denser spacing to illustrate what can be achieved with this method. We have therefore updated the manuscript to show a full array of 120 printed devices printed with a pitch of X and Y μm in the x and y directions respectively.

“The wavelength ordered set of PhCCs presented above was printed with minimum pitch of 50 μm in both the x and y directions. In order to demonstrate the potential of this integration method for producing densely packed arrays we transferred an additional 120 devices onto an 8 \times 15 spatial array of support frames with a pitch of 18 and 16 μm in the x and y directions and with a minimum edge to edge separation of 7 μm in both directions, as shown in Fig. 7.”

7. *Please give explanations why wavelength shifts of PhCCs are alleviated after 2nd print (Fig. 5).*

AND

8. *Please give more detailed explanation why the PhCC after transfer has red-shift of resonant wavelength, not blue-shift (Fig. 7).*

In response to the reviewer's points 4,7 and 8, we have now added more detail on the mechanical effects of the transfer printing process, specifically addressing the effects of the initial tether fracture and release of the membranes, compared with site to site printing.

In answer to point 7, the permanent wavelength shift in the cavities induced during the first print is correlated with the mechanical fracture of the tether as a source of strain in the material and hence effective refractive index shift of the cavity mode. Further detail is provided in response to point 4 with regards to the magnitude of these effects.

In answer to point 8, the redshift measured during the dynamic measurements is due to an elastic effect that relaxes over the measurement timescales back to the steady-state cavity resonance wavelength that is consistent with the pre-print characterisation. As noted in response to point 4, the most likely cause of the cavity wavelength shift is the mechanical deformation of the cavity, resulting in an effective spatial increase in the photonic crystal lattice that results in a wavelength shift from the steady-state cavity case. The red-shift of the cavity resonance would be correlated with an increased lattice constant of the order of 0.56 nm. The optical measurement of the cavity resonance is the most sensitive metrology tool we have available and other imaging techniques (for example Scanning Electron Microscopy) would not be able to resolve this scale of modulation in the time-frames available. The fact the cavity redshifts rather than blue shifts is likely due to the forces created when the PhCC is brought into contact with the suspension frames which put it into extension rather than compression, likely due to the shear motion used in the release process. Aside from the optical measurements the cause of this direction of force is difficult to assess, though may be due to the deflection of the cavity as it is brought into contact that then relaxes into a flat steady state membrane.

These considerations have now been added to the manuscript discussion section.

Reviewer 2

1. *The authors have shown that the TP process can transfer a single pixel element up to five times before the shift in resonance wavelength exceeds the line width, but it seems that the overlap with the receiver area of less than 1 micron can be applied under special constraints such as the current size of the PhCC and the slab thickness so on. I would like to suggest the authors to show the dependence of the success rate of TP on the overlap area with the receiver area and the dependence of the PhCC slab thickness, etc.*

The reviewer is correct that the spatial overlap of the PhCC with the air gap void area is crucial to maintaining the performance of the device in terms of resonant wavelength and cavity Q-factor. If the area of the PhCC area is in contact with the suspension frame this would reduce the index contrast in that area and modify the device optical performance. For that reason, we designed the membrane devices to include a planar area around the PhCC device to isolate the PhCC from the suspension frames underneath. The design trade-off in the definition of this planar area was, as suggested by the reviewer, the balance between having enough contact area to create a stable bond between the membrane device and the suspension frame, and minimising the potential pitch between adjacent PhCC's to maximise active optical area.

The definition of minimum contact area for successful printing of devices is explored in the literature, including demonstrations of micron-sized LED pixels [Gomez et al., Proc. Elec. Comp. and Tech. Conf., 2017], nanowire devices [Jevtics et al., Appl Phys Rev. 9(4), 2022], and III-N micro-lenses [Wessling et al., Opt. Mat. Exp., 12(12), 2022]. The yield of the process with 1um planar area edges for this work was 100% of releasable devices. Furthermore, this is not dependent on other conditions including the thickness of the slab:

“The vast majority of work has focussed on the printing of membrane devices onto planar substrates where contact is made across the full membrane surface [4, 30–33]. Transfer of devices with surface contact dimensions in the few micron range have been demonstrated with high-yield and repeatability for full surface contact of micro-LED pixels [34, 35], micro-lenses[36], and nanowires [37]. Successful transfer printing has also been demonstrated for devices where the contact area of printed devices is in excess of any suspended regions or areas with topological distortion [38–40].”

Furthermore, we have now added further detail on the transfer printing process and demonstration of printed device overlap as detailed in response to Reviewer 1's points 1 and 5.

2. *The TP equipment developed by the authors is excellent but appears to be a standard one, and it is difficult to see its originality. For example, the swept wavelength spectrum measurement system integrated with the TP equipment does indeed enable in-situ evaluation of optical properties, but it does not seem to have anything particularly new in terms of scientific development. I would like suggest the authors to emphasize the novelty of the TP device to be clearly.*

We agree that the novelty of our instrument was not sufficiently emphasised in the original manuscript. Please also see the response to reviewer 1, point 2. Although transfer printing and optical spectrum analysis of vertically coupled cavities are both established technologies, the integration of these into a single toolset is non-trivial. Another important addition to our system

is the use of a broadband, high spatial resolution camera for the imaging and spectral measurement functions of the system. By using a camera with sensitivity through the visible range of the spectrum up to the near-IR telecommunications bands we can image the cavities for mechanical transfer and use the camera sensor for the picometer resolved swept laser spectral measurements and cavity mode reflectivity imaging. Under widefield excitation, the setup would also allow for parallel excitation and readout of numerous cavities. We have now highlighted these unique advances more clearly in the manuscript. The substantial advance in throughput of device transfer, and the measurement of mechanical dynamics, which the integration of transfer and measurement systems enables, represents significant novelty over the current state-of-the-art in 'blind' transfer printing systems. Existing systems can only deal with the mechanical structures observable via optical imaging systems, and require the use of post-integration measurement stages and the rework that these would entail if devices were not as specified for the printing selection.

In summary, the major novelty of our work, dependent on the custom-built system, is:

- **The integration of optical spectral measurements, visible and IR imaging of PhCCs in the same system that allows high-accuracy transfer printing of the devices.** This enables rapid measurement and transfer of large numbers of devices with spatial positioning accuracy in the 100's of nanometres range. We present the measurement, binning and transfer of 119 devices in a single session of a few hours here. The bottleneck in this demonstration is the serial measurement of the PhCCs' spectra which is in the order of 2 minutes per measurement. This could be significantly reduced by moving to a wide field of view, parallel measurement of the cavity resonances.
- **Micro-assembly of arrays of PhCCs ordered by resonant wavelength.** This demonstrates the breaking of the limitation of fabrication disordering of nominally identical devices. As noted in response to Reviewer 2 point 1 above, the ability to transfer compact membrane devices with a support planar region of only 1um in dimension is crucial for this form of arrayed, suspended devices. The automated transfer of this form factor of devices requires the 6-axis, velocity and spatial vector controlled printing processes now discussed explicitly in the manuscript, as detailed in response to Reviewer 1 point 1.
- **The measurement of the dynamic effects of transfer printing** are presented for the first time, show the changing cavity response in the seconds to hours timescale after mechanical printing. This is only possible through the integrated measurement and transfer tool we have constructed as detailed above.

We have highlighted these elements in the updated abstract and throughout the manuscript.

- 3. The dynamic effect of the TP on the resonance wavelength of the cavity is measured on a time scale of several seconds to several hours, and the elastic effect of the response is discussed, which is an interesting result, but the physics that underlies this time scale is not clear. In order to enable more comprehensive discussion, the authors should show the dependence of the closeness of the receiver and the PhCC, the overlapping area, the PhCC slab thickness, and the overall size, etc.*

See answer to Reviewer 1 point 4. We have also included a new discussion section to the manuscript focussing on the magnitude of the cavity optical resonance shifts and the likely physical origins.

4. *The TP technology developed in this study is a combination of existing technologies, and its usefulness of PhCCs array as a device development has not been demonstrated, so it remains at the stage of half-baked research. The authors are expected to clarify how the array device ordered along the wavelength will be used in practice.*

We thank the reviewer for highlighting this aspect of the manuscript. We agree that the implications of the demonstrations we present could have been better elucidated and that some discussion of the impact of this work on the field should be presented. We have now included a discussion section to the manuscript to better highlight the novelty and substantial advances on the state-of-the-art made by this work and to detail the implications for future systems, in particular in the integration of compact PhCC membrane devices on non-native substrates.

“The sub-nanometre wavelength shifts of all cavity steady-state resonances observed after 5 sequential printing cycles demonstrates the ability to preserve all physical properties of the device (lattice constant, refractive index, etc.) to better than one part in 1000. The preservation of resonant cavity wavelength allows for the selection of specific cavities to be integrated into pre-defined spatial locations, overcoming the existing variance in nominally identical devices as-fabricated on their native substrate. This deterministic post-measurement selection of cavities will enable single device selection for integration into photonic integrated circuits, or on fiber tips, as high-quality factor filters and non-linear resonators with absolute control over target wavelength. Importantly, the spatial ordering of these devices enables construction of cavity arrays with well-defined wavelength characteristics. For example, cavity arrays can be designed with wavelength gradients across the array to enable passive beam steering at high speeds [47]. Uniform arrays can also be assembled for applications as spatial light modulators [20]. In this case the yield of cavity responses within the linewidth scale is important and is related to the variance of cavity wavelength from nanofabrication tolerances. In this work we show that groupings of ≈ 5 PhCCs can be extracted within the range of a linewidth from an initial set of 120 devices. Full details are presented in the supplementary material. This gives a nominal down selection factor of ≈ 0.042 , meaning that for a uniform 10×10 cavity array, an initial donor set of around 2380 devices is needed. Complementarily, for donor arrays at large scale, multiple uniform arrays can be assembled from a single set of measurements.”